# DIFFERENTIABLE SENSOR LAYOUTS FOR END-TO-END LEARNING OF TASK-SPECIFIC CAMERA PARAMETERS

## ABSTRACT

Computational imaging concepts based on integrated edge AI and neural sensor concepts solve vision problems in an end-to-end, task-specific manner, by jointly optimizing the algorithmic and hardware parameters to sense data with high information value. They yield energy, data, and privacy efficient solutions, but rely on novel hardware concepts, yet to be scaled up. In this work, we present the first truly end-to-end trained imaging pipeline that optimizes imaging sensor parameters, available in standard CMOS design methods, jointly with the parameters of a given neural network on a specific task. Specifically, we derive an analytic, differentiable approach for the sensor layout parameterization that allows for task-specific, locally varying pixel resolutions. We present two pixel layout parameterization functions: rectangular and curvilinear grid shapes that retain a regular topology. We provide a drop-in module that approximates sensor simulation given existing high-resolution images to directly connect our method with existing deep learning models. We show for two different downstream tasks, classification and semantic segmentation, that network predictions benefit from learnable pixel layouts. Moreover, we give a fully featured design for the hardware implementation of the learned chip layout for a semantic segmentation task.

## 1 INTRODUCTION

Deep learning models have achieved impressive performance in a wide range of computer vision tasks, including image classification, object detection, and semantic segmentation. Computational imaging approaches go beyond the traditional paradigm in computer vision, which uses an image as input, and jointly considers the image processing method and the image formation process performed in a camera, comprising various hardware design and low-level, on-chip image processing parameters. Recent approaches in deep computational imaging aim at a task-specific optimization of image signal processing pipeline (ISP) parameters (Mosleh et al., 2020), various optical components (Chang et al., 2018; Chang & Wetzstein, 2019; Chugunov et al., 2021; Metzler et al., 2020; Sun et al., 2020; Tseng et al., 2021), or color filter layouts (Chakrabarti, 2016) in tandem with neural network parameters in an end-to-end fashion.

While the trend toward high-resolution imagery remains unbroken, specifically mobile systems that incorporate computer vision solutions are strongly focused on cost, energy, and resource efficiency, as well as on data security. Therefore, another trend in deep computational imaging addresses new hardware concepts that integrate edge AI and neural sensor concepts (Iturbe et al., 2023; Suo et al., 2021; Martel & Wetzstein, 2021; Klinghoffer et al., 2022). These approaches also solve vision problems in an end-to-end, task-specific manner, jointly optimizing the vision algorithm and the imaging hardware parameters (Klinghoffer et al., 2022). Here, the primary goal is to sense data with high information value useful for the final task (Suo et al., 2021; Iturbe et al., 2023), thus minimizing energy and data communication requirements as well as the amount of captured data to maximize privacy. While the integration of AI compute and sensing capabilities is a mid to long term hardware development trend (Iturbe et al., 2023), it results in reduced amount of visual data acquired without losing task performance. An example is the in-pixel-processing with a SCAMP-5 chip with $256^2$ px (Martel & Wetzstein, 2021).

Our work is motivated by the hardware trend towards integrated AI compute and sensing capabilities (Iturbe et al., 2023), but we investigate the option to use currently available parameters in standard CMOS sensor design processes. We specifically focus on non-uniform and minimalistic pixel layouts, i.e., the number, location, size, and, potentially, shape of pixels on the image sensor. This objective is in line with the common practice to artificially decrease the image resolution at the input for network training to adhere to memory constraints of the GPU. We make the flexibility in pixel layout design accessible to deep learning, and propose a data-driven approach that optimizes the pixel layout for a given task together with the neural network parameters in an end-to-end fashion. We demonstrate, that various tasks, such as autonomous driving, benefit from specifically optimized pixel layouts, especially if they induce a *spatial bias*, where certain image regions require a higher pixel density than others regions. To the best of our knowledge, this is the first approach for joint optimization of pixel layout and downstream tasks which is realizable by current hardware limitations of sensors. We validate the effectiveness of our method on classification and semantic segmentation tasks and demonstrate a significant improvement over baselines without significantly depending on the specific network utilized to solve the task. Our contributions are as follows:

- We propose a differentiable, physically based sensor simulation framework, that allows end-to-end gradient-based optimization of pixel layouts,
- A generic pixel layout parameterization that covers a large class of possible geometries, including rectangular and free-form pixel shapes, while retaining the regular topology, usually required by downstream networks, and
- A drop-in module, that can approximate the sensor simulation given existing high-resolution images and can thus be easily incorporated into existing deep-learning models.
- We show experimentally that tasks like semantic segmentation in autonomous driving can benefit from non-uniform pixel layouts, and
- We give a fully featured chip design that realizes the learned chip layout for semantic segmentation in autonomous driving.[1]

## 2 RELATED WORK

In this section, we discuss prior work that motivates the need for non-uniform *pixel layouts*, attempts to *end-to-end optimization of ISP pipelines* for various downstream tasks involving optimizing sensor parameters for image quality or learning effective downsampling, as well as *superpixel-based methods* that end-to-end optimize superpixel generation and network parameter optimization.

**Pixel Layout.** In computer vision tasks like semantic segmentation, most methods employ an "hour-glass model" like UNet (Ronneberger et al., 2015), PSPNet (Zhao et al., 2017), etc. to be computationally efficient. Here, the information is first encoded as low-resolution feature maps and then upsampled to image resolution for the downstream task at hand. However, this approach makes the strong assumption that all regions in the image have equal information, thus equal importance. In practice, this assumption does not hold as Jin et al. (2021) show that a network's performance can benefit from non-regular, learnable downsampling strategies. Jin et al. (2021), however, use a fixed, high-resolution input image of 2Mpx and above for training **and** inference, whereas we aim at optimizing the **lowres input** sensor layout for inference. Additionally, some early works argue for non-regular pixel layouts due to their ability to improve super-resolution (Ben-Ezra et al., 2007) or to provide better image representations as such (Kirsch, 2010).

**End-to-end Optimization of the ISP pipeline.** Some attempts have been made to jointly optimize the ISP pipeline and network parameters (Mosleh et al., 2020; Jin et al., 2021; Marin et al., 2019; Talebi & Milanfar, 2021). In their work, Mosleh et al. (2020) proposed a "hardware-in-the-loop" method to jointly optimize the hardware ISP and network parameters as a multi-objective problem using a $0^{th}$-order stochastic solver for perceptual image quality. Other prior work account for the incident pixel radiance in an end-to-end optimization, which we also optimize to learn the sensor's pixel layout. Talebi & Milanfar (2021) use a "CNN-based resizer" to downsample high-resolution images and then perform classification using deep learning-based recognition models and jointly learn weights for the "resizer" and recognition model, which is extremely complex, is not buildable as a sensor, and only performs uniform downsampling. Marin et al. (2019) proposed an edge-based downsampling scheme that maps images to a non-uniform pixel layout with a focus on

---

[1]The manufacturable chip with non-uniform pixel layout is in the process of being submitted for fabrication.

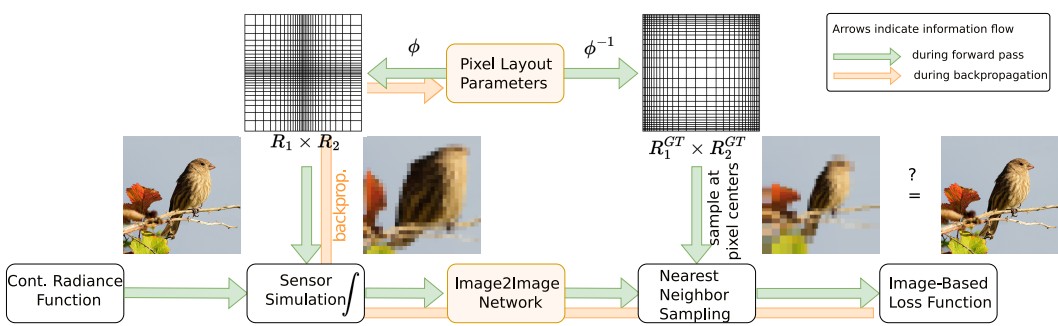

Figure 1: The overall pipeline for end-to-end learning of task-specific pixel layouts. $R_1 \times R_2$ and $R_1^{GT} \times R_2^{GT}$ denote the resolution of the sensor and reference image resolution, respectively.

object boundaries, i.e., edges. They learn to sample with a higher resolution near object boundaries and with lower resolution elsewhere. This helps in improving the computation cost while retaining some task-essential information. Nonetheless, many tasks also require information from the object's interior, e.g., cancer cell detection. To overcome this drawback, Jin et al. (2021) proposed to learn a deformed sampling density distribution used for downsampling a high-resolution image over a non-uniform grid based on the network's performance on the downstream task rather than the edge vicinity. However, Jin et al. (2021); Marin et al. (2019) use a neural network *on the high-resolution image* to predict a suitable pixel layout for each image individually, requiring the neural network to operate on high-resolution data. On the contrary, our approach learns the hardware sensor layout with the subsequent network operating on the produced low-resolution data only, considering the full pixel radiance, not just a single sampling position. Our drop-in module can take existing high-resolution images to approximate the incident radiance, allowing to optimize any network for a limited resolution budget or to reduce the network's size and inference time while keeping performance as high as possible. In prior work Riad et al. (2022), similar approaches have been taken by *uniformly* decreasing pixel resolution of the input images to reduce the network size until the performance falls below a given lower bound. Our approach can thus be seen as a generalization of Riad et al. (2022), integrating non-regular sampling schemes like Jin et al. (2021), while considering the total intensity of the input image (as an approximation of the incident radiance to a sensor).

**Superpixel-based Methods.** Apart from learning sensor parameters for effective downsampling, other approaches learn pixel grouping into larger superpixels and end-to-end optimizing to learn this grouping and network parameters (Yu & Fan, 2021; Fey et al., 2018; Avelar et al., 2020). Yet, all such approaches create *image specific* superpixels and are therefore not suitable for learning a *fixed* image sensor layout. Moreover, as inspired by Chang et al. (2018); Chang & Wetzstein (2019); Chugunov et al. (2021); Metzler et al. (2020); Mosleh et al. (2020); Sun et al. (2020); Tseng et al. (2021) our objective rather is to optimize sensor parameters (or image downsampling) in a photometrically correct way, i.e., considering the continuous acquisition of radiance at the pixel level. Another related technique is hardware pixel binning (Zhang et al., 2018; Yoo et al., 2015; Westra et al., 2009; Cho et al., 2014; Mennel et al., 2022), e.g., for reducing noise in low-light situations. Commonly, these approaches apply uniform binning. While pixel binning can be used to implement our non-uniform layouts in FPGA and allow modifications of the pixel layout without re-manufacturing the sensor, the hardware resource and energy requirements remain unchanged. We, therefore, designed a chip with an optimized pixel layout and plan to build it directly in silicon.

## 3 DIFFERENTIABLE SENSOR SIMULATION

As opposed to usual deep learning techniques, which pick a task-specific network architecture $\mathcal{G}$ to make predictions $\mathcal{G}(I, \nu)$ on input images $I$ for parameters $\nu$ of the network, we propose to consider the images $I$ as functions $I(\theta, L_i)$ of the incoming radiance $L_i$ as well as a parameter vector $\theta$ that describes the diffeomorphic deformation of the pixel layout the image $I$ is recorded with. This naturally requires modeling a spatially continuous sensor simulation process that determines the value of each pixel for a given deformation described by $\theta$. As soon as the sensor model, i.e., the dependence of the recorded image $I$ as a function of the deformation parameters $\theta$ and the radiance $L_i$ is determined, we propose to *jointly* train our system for network as well as sensor

system parameters by optimizing

$$\min_{\theta,\nu} \mathbb{E}_{(L_i,y)}(\mathcal{L}(\mathcal{G}(I(\theta,L_i),\nu),y)), \tag{1}$$

where $\mathcal{L}$ is a suitable loss function to compare the network's prediction to the desired prediction $y$.

Our overall approach for representing $I$ as a function $I(\theta, L_i)$ is depicted in Fig. 1. Assuming a *continuous radiance function* (or a high resolution image) $L_i \in \mathbb{R}^{R_1^{GT} \times R_2^{GT}}$ as input, we perform a *sensor simulation* (see Sec. 3.1) to capture the radiance hitting the individual pixels defined by our *pixel layout parameters* (see Sec. 3.2). Our pixel layout is a continuous, bijective function $\phi$ applied to a regular pixel grid that retains the pixel topology. Thus, the resulting, usually distorted pixel layout can be fed to any network that accepts images as input. In case of a downstream image-to-image network that uses an image-based loss function, we apply a *back-warping*, i.e. an interpolation using $\phi^{-1}$, to resample the network's output to the original image resolution. In case of a classification network, no back-warping is required. As our pixel layout model is differentiable, the optimal task-dependent pixel layout can be learned jointly with the network (see Sec. 3.3).

## 3.1 MEASUREMENT EQUATION

Here we present the image formation process, specifically the measurement on the sensor plane. We denote pixels with the multi-index $k = (k_1, k_2)$. An individual sensor pixel measures the energy $E_k$ of incoming radiance $L_i$, integrated over all possible directions $\omega \in \Omega \subseteq \mathbb{R}^2$, locations $p \in A_k \subseteq \mathbb{R}^2$, time $t \in [t_0, t_1]$ inside the exposure window and measurable wavelengths $\lambda \in [\lambda_0, \lambda_1]$:

$$E_k = \int_{\lambda_0}^{\lambda_1} \int_{t_0}^{t_1} \int_\Omega \int_{A_k} W(p,\omega,t,\lambda)L_i(p,\omega,t,\lambda)dpd\omega dtd\lambda. \tag{2}$$

Here, $W$ is a weighting term that models the varying responsivity of the sensor w.r.t. the parameters, like the location inside a pixel or sensitivity to specific wavelengths. The measured energy is then further processed in an image signal processing pipeline (ISP) to compute an RGB value $I_k \in \mathbb{R}^3$.

For simplicity, we assume a static scene that is captured through a pinhole camera. Further, we interpret the radiance function to be independent of the wavelength and instead output RGB values. This allows us to directly compute $I_k$ as the mean RGB value over the pixel area:

$$I_k = \frac{1}{\text{vol}(A_k)} \int_{A_k} W(p)L_i(p)dp \tag{3}$$

Note, that it is often (implicitly) assumed that each pixel has unit area, ignoring the factor in front of the integral. Since we want to model pixels of different sizes, we have to keep the normalization factor. Generally, this normalization is part of the ISP that maps energies to RGB values.

## 3.2 PARAMETERIZING PIXEL LAYOUTS

In the following, we introduce a general framework to parameterize the pixel layouts. We set the full sensor area to be the square region $\mathcal{S} = [-1, 1]^2$. As such, each pixel has its own domain $A_k \subset \mathcal{S}$ and weighting function $W_k$. We assume pixels to be disjoint, except for their boundaries $\partial A_k$, and their union to cover the whole sensor area, i.e., $\mathcal{S} = \bigcup A_k$. We denote pixels from the standard, uniform layout as $U_k$. In this case, the pixel boundaries match the parallel grid lines of the sensor, i.e., for the sensor resolution $R_1, R_2$,

$$U_k = \left[\frac{2k_1 - R_1}{R_1}, \frac{2(k_1+1) - R_1}{R_1}\right] \times \left[\frac{2k_2 - R_2}{R_2}, \frac{2(k_2+1) - R_2}{R_2}\right]. \tag{4}$$

We now define a class of pixel layouts to be a parameterized deformation function $\phi : \mathcal{S} \times \mathcal{D} \rightarrow \mathcal{S}$, where $\mathcal{D} \subset \mathbb{R}^d$ is the set of possible parameters. We require the function $\phi(\cdot, \theta)$ to be bijective and bi-Lipschitz, implying that $\phi$ and $\phi^{-1}$ are differentiable almost everywhere for fixed $\theta \in \mathcal{D}$. We define the pixels under this layout as $A_k(\theta) = \phi(U_k, \theta)$, i.e., the image of the uniform pixel area. The bijectivity constraint assures that the deformed pixels do not overlap and the overall pixel number and topology is retained. Since $\phi$ is Lipschitz, neighboring pixels are mapped to neighboring

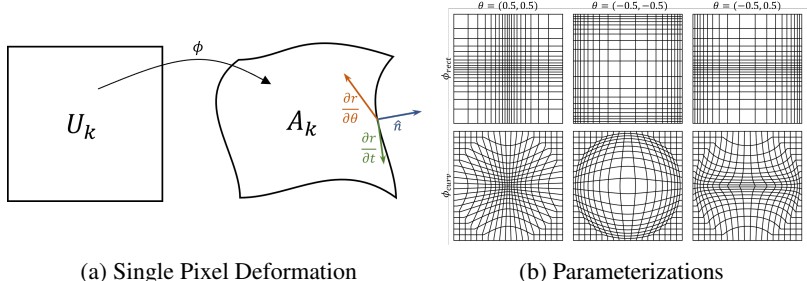

(a) Single Pixel Deformation                  (b) Parameterizations

Figure 2: (a) Applying $\phi$ to a uniform pixel $U_k$ yields the deformed pixel $A_k$. The flux integral for gradient computation requires quantities on each boundary point $r \in \partial A_k$: The tangent $\frac{\partial r}{\partial t}$, the outward-pointing normal $\hat{n}$, and the change of the boundary point with respect to the pixel layout parameterization $\frac{\partial r}{\partial \theta}$. (b) The rectangular (top row) and curvilinear (bottom row) pixel layouts used in this paper and their behavior under different parameters on a $20 \times 20$ sensor.

pixels again and boundaries get mapped to boundaries, i.e., $\partial A_k(\theta) = \phi(\partial U_k, \theta)$. The forward pass of the image formation process, i.e. the pixel color is computed by a change of variables

$$I_k(\theta) = \frac{1}{\text{vol}(A_k(\theta))} \int_{A_k(\theta)} W_k(p, \theta) L_i(p) dp = \frac{\int_{U_k} W_k(\phi(u, \theta), \theta) L_i(\phi(u, \theta)) |\det J_\phi(u, \theta)| du}{\int_{U_k} |\det J_\phi(u, \theta)| du}$$
(5)

Here $J_\phi$ is the $2 \times 2$ Jacobian of $\phi$ with respect to the spatial arguments. Since the weighting function may depend on the pixel layout, we explicitly added $\theta$ to its arguments.

Since rectangular pixels have an advantage in manufacturing the optimized layout in hardware, we propose a simple deformation, that depends only on two parameters. Still, in our experiments, we found that even such a simple pixel layout already leads to significant improvements. We consider the deformation function given $p = (p_1, p_2) \in \mathcal{S}$ and $\theta = (\theta_1, \theta_2) \in (-1, 1)^2$, i.e.

$$\phi_{rect}(p, \theta) = \begin{pmatrix} \frac{p_1(\theta_1 - 1)}{2\theta_1 |p_1| - \theta_1 - 1} \\ \frac{p_2(\theta_2 - 1)}{2\theta_2 |p_2| - \theta_2 - 1} \end{pmatrix}$$
(6)

The parameters control the vertical and horizontal deformation strengths and, dependent on their sign, move more pixels to the edges or the center of the sensor.

Currently, there exist no comprehensive hardware design constraints for pixel layout, but designs beyond rectilinear ones are possible. We, therefore, exemplarily show that our method can also handle curvilinear layouts with a circular version of Eq. (6) (see Fig. 2b, namely

$$\phi_{curv}(p, \theta) = \begin{pmatrix} \frac{p_1(\theta_1 - 1)}{2\theta_1 ||p||_2 - \theta_1 - 1} \\ \frac{p_2(\theta_2 - 1)}{2\theta_2 ||p||_2 - \theta_2 - 1} \end{pmatrix} \quad \text{if } ||p||_2 < 1, \qquad \phi_{curv}(p, \theta) = \begin{pmatrix} p_1 \\ p_2 \end{pmatrix} \quad \text{otherwise.} \quad (7)$$

### 3.3 End-to-End Optimization

Since the goal is to optimize the pixel layout for some downstream task, we will now explain how to incorporate our pixel layout parameterization in a gradient-based optimization scheme. The pixel-outputs of the sensor simulation $I_k(\theta)$ (see Eq. (5)) are processed by a neural network $\mathcal{G}$. We compute the gradient of a task-specific loss function $\mathcal{L}$ with respect to the sensor parameter $\theta$ by applying the chain rule

$$\frac{\partial \mathcal{L}}{\partial \theta_j} = \sum_k \frac{\partial \mathcal{L}}{\partial I_k} \frac{\partial I_k}{\partial \theta_j},$$
(8)

where $\frac{\partial \mathcal{L}}{\partial I_k}$ can be computed via standard backpropagation.

In the following, we give an analytic expression for the second term, i.e. the derivative of a pixel color with respect to the sensor layout. For brevity, we will drop the $j$ subscript so that $\theta$ always refers to a scalar value. However, the calculations can be easily vectorized.

Starting from Eq. (5), applying the quotient rule yields

$$\frac{\partial I_k}{\partial \theta} = \frac{\partial}{\partial \theta} \left( \frac{1}{\text{vol}(A(\theta))} \int_{A_k(\theta)} W_k(p,\theta) L_i(p) dp \right) =: \frac{\partial}{\partial \theta} \frac{f(\theta)}{g(\theta)} = \frac{\frac{\partial f}{\partial \theta}(\theta) g(\theta) - f(\theta) \frac{\partial g}{\partial \theta}(\theta)}{g(\theta)^2}. \quad (9)$$

We explain the derivative of $f$, i.e., $\frac{\partial f}{\partial \theta} = \frac{\partial}{\partial \theta} \int_{A_k(\theta)} W_k(p,\theta) L_i(p) dp$.

The derivative of the volume can be computed analogously. Since the integration domain itself depends on $\theta$, we can apply Reynold's transport theorem (Flanders, 1973):

$$\frac{\partial f}{\partial \theta} = \int_{A_k(\theta)} \frac{\partial}{\partial \theta} W_k(p,\theta) L_i(p) dp + \oint_{\partial A_k(\theta)} W_k(r,\theta) L_i(r) \langle \frac{\partial r}{\partial \theta}, \hat{n}(r,\theta) \rangle ds$$
$$=: Q_{int}(\theta) + Q_{bound}(\theta) \quad (10)$$

The first integral can be easily computed by a change of variables like Eq. (5). The second integral is the flux integral across $\partial A_k$, where $r : [0,1] \to \partial A_k(\theta)$ is an arbitrary piece-wise smooth parameterization of the boundary, $\frac{\partial r}{\partial \theta}$ is the local change of the boundary point with respect to $\theta$ and $\hat{n}$ is the corresponding outward pointing unit normal vector. Since the boundary of $A_k$ is the image of the boundary of $U_k$ under $\phi$, a parameterization can be given by $r(t,\theta) = \phi(\gamma(t),\theta)$, where $\gamma$ is a parameterization of the square boundary of $U_k$. Furthermore, tangent vectors of $r$ are the pushforward of tangent vectors of $\gamma$ by $\phi$, i.e., $\dot{r}(t,\theta) = J_\phi \dot{\gamma}(t,\theta)$. Thus, we can compute the line element $ds = ||\dot{r}(t,\theta)|| dt$ and the normal (up to orientation) as

$$\hat{n}(t,\theta) = \begin{pmatrix} \dot{r}_2(t,\theta) \\ -\dot{r}_1(t,\theta) \end{pmatrix} / ||\dot{r}(t,\theta)||_2. \quad (11)$$

The important quantities of the boundary integral are visualized in Fig. 2a. With this, we have all ingredients to compute the derivative using Eq. (10). The interior integral $Q_{int}$ looks very similar to the pixel color formula in Eq. (5) and can be efficiently computed by reusing radiance samples from the forward pass. The boundary integral $Q_{bound}$ only requires sampling the border of the pixel, which requires less samples, which can even be reused for neighboring pixels. Note, that applying the divergence theorem on $Q_{bound}$ would transform it into another interior integral. However, in that case we would also need to compute the gradients of $L_i$ with respect to the location on the sensor plane. Using the formulation above, we do not require the availability of gradients of $L_i$.

## 4 IMPLEMENTATION DETAILS

**Simulation framework.** We implement both, the forward and backward pass, using numerical integration. We employ stratified Monte-Carlo integration to reduce possible aliasing artifacts that could emerge from quadrature-based integration schemes. We make no assumptions on how $L_i$ is calculated. In general, our method can be built on top of any existing rendering algorithm, as long as it exposes a way to sample the radiance at given points on the image plane.

The main goal of our method is to enhance a downstream deep learning task, which generally requires a substantial amount of training data. For many tasks, large and widely-used datasets are available, whose effectiveness has been proven over time. We therefore decided to approximate the radiance function $L_i$ with high-resolution real images, which we transform into a coarser image using our differentiable sensor simulator. We use bilinear interpolation to sample the radiance at arbitrary positions on the sensor plane. Furthermore, we set $W_k$ to be constant, i.e. all points on a pixel have the same sensitivity.

We feed the output of the sensor simulation directly to the downstream network, as if it were on a uniform grid (see Fig. 1). In case the downstream network is solving an image-to-image task the output of the network needs to be resampled in order to match the reference image pixel grid. We give details on that case, as well as general implementation details in Appendix A.

**Sensor Hardware.** We give an overview of the general approach in designing rectilinear pixel layouts using the learned layout for the specific application of semantic segmentation based on the Cityscape dataset, see Fig. 4b. Note, that the sketched design approaches can, e.g., also be applied to smaller base pixels, while non-rectilinear layouts require different procedures.

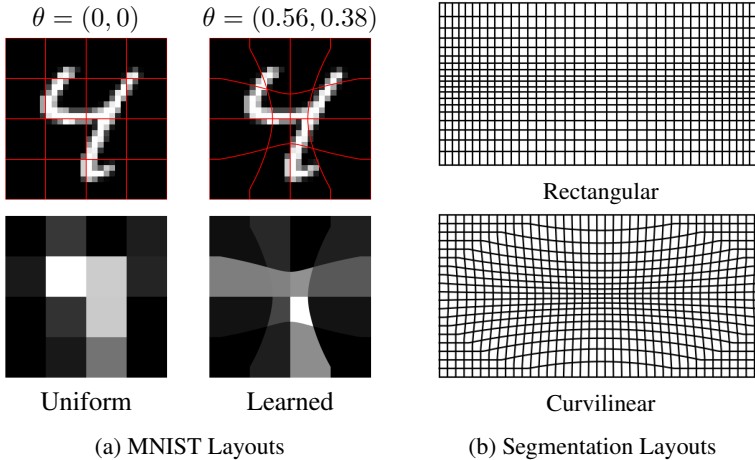

$\theta = (0, 0)$      $\theta = (0.56, 0.38)$

Uniform     Learned           Curvilinear

(a) MNIST Layouts      (b) Segmentation Layouts

Figure 4: (a) Learned sensor layout for $4 \times 4$ MNIST classification. Top row: pixel layouts overlayed over the original $28 \times 28$ image. Bottom row: Simulated sensor output. The network fine-tuned on the uniform layout wrongly classifies the image as a 7 while the end-to-end optimized network classifies as a 4. (b) Learned rectangular and curvilinear pixel layouts for $256 \times 128$ segmentation on Cityscapes with PSPNet (less pixels are shown for visualization purposes).

We designed and laid out the image sensor using an $0.18\mu$m XFAB CMOS image sensor process. We select a typical 4-transistor active pixel setting with a minimal transistor size of $0.35\mu$m $\times$ $0.22\mu$m, and a base pixel of $5\mu$m $\times$ $5\mu$m. Fig 3 depicts the layout comprising photodiode (red), the power supply (blue), the output line (yellow), and the reset and transfer gate (white), all shielded by a top metal layer. The base pixel positions are rounded off to meet $0.18\mu$m manufacturability, and the "free space" for larger pixels is filled with photodiodes. The



Figure 3: Base pixel (left) and non-uniform (right) IC layouts for the sensor optimized for segmentation in Sec. 5.

smallest and largest pixel are $6.534\mu$m$\times5.76\mu$m and $17.91\mu$m$\times10.26\mu$m, respectively. The fill factor, which determines optical efficiency, without microlenses for the base pixel is 57%. It rises to 61.2% for the smallest pixel and to 83.42% for the largest pixel in the array. The entire array size is 2mm$\times$1.4mm, which is fairly small. We expect a very small reduction in the overall intensity dynamic range due to the varying pixel size, which can be counteracted using a modified dual sampling technique allowing for adaptive integration times. For further details, we refer to the supplementary material.

The chip will be submitted for fabrication using XFAB's XS018. The next downstream stages to obtain an operative prototype camera comprise the finalization of the design, including its interface circuits and tape-out, as well as testing of the chip's functionality and the integration into a camera.

## 5 EXPERIMENTS

We conduct several experiments both for classification and semantic segmentation. For each experiment, we compare a task-specific network with uniform sensor layout as the baseline to the same network end-to-end trained with a learned sensor layout, which is fixed during inference and evaluated on simulated data. Details on all training schemes and used network architectures can be found in Appendix B.

**MNIST classification.** To illustrate the principle of our end-to-end sensor layout optimization, we start with a toy example and optimize the layout for hand-written digit recognition on MNIST (Deng, 2012) with a sensor size of only $4 \times 4$ instead of the original $28 \times 28$ pixels. The digits in MNIST

| Network | PSPNet | | | SegNeXt | | | DeepLabV3plus | | |
|---|---|---|---|---|---|---|---|---|---|
| Layout | Uniform | $\phi_{curv}$ | $\phi_{rect}$ | Uniform | $\phi_{rect}$ | $^\dagger\phi_{rect}$ | Uniform | $\phi_{rect}$ | $^\dagger\phi_{rect}$ |
| mIoU | 50.76 | 52.39 | **54.08** | 49.16 | 50.01 | **51.07** | 55.13 | **56.68** | 55.95 |
| Acc. (%) | 90.86 | 91.19 | **91.44** | 90.12 | 90.22 | **90.52** | 91.61 | **91.83** | 91.79 |

Table 1: Semantic segmentation results on Cityscapes dataset (Cordts et al., 2016) ($256 \times 128$). Fixed layout transferred from the optimal layout of PSPNet is indicated by †. Otherwise layouts were trained from scratch. Bold indicates the best result for that network architecture.

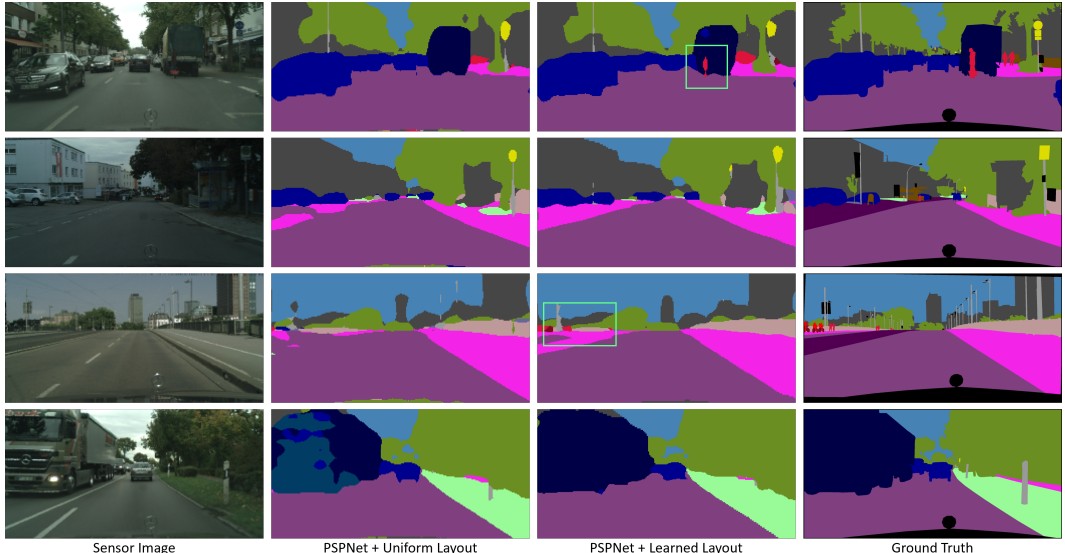

Figure 5: Example segmentations from the Cityscapes (Cordts et al., 2016) test set with and without learned sensor layouts at $256 \times 128$ resolution. The used deformation is $\phi_{rect}$. To accommodate the different pixel shapes, both variants were upsampled using nearest neighbor interpolation to the ground truth resolution ($2048 \times 1024$), as described in Appendix A. The learned pixel layout achieves more accurate segmentations with objects detected that were otherwise missed by using the uniform layout, especially in the vertical center of the image.

are always centered, so the hypothesis is that an optimized layout puts smaller pixels in the middle in order to capture the higher information density. This is in contrast to more general datasets like ImageNet (Deng et al., 2009), which aim to provide images from a distribution that is as wide as possible and thus have little to no spatial bias. We apply the curvilinear layout $\phi_{curv}$ whose parameters we initialize with 0, i.e. we start the training with a uniform pixel layout.

Overall, the learned layout (Fig. 4a) achieves a testset accuracy of 90.90% and the uniform layout only 83.63%. The layout confirms our hypothesis that smaller pixels in the center are advantageous for classification on this specific dataset. Note that the visualization of the learned layout in Fig. 4a is not what the network receives as input. The CNN has no notion of pixel shape and sensor layout, and only "sees" a $4 \times 4$ grid of pixel values.

**Semantic segmentation.** For a relevant practical example application, we evaluate the learned pixel layouts under the task of semantic segmentation of urban street scenes on the Cityscapes dataset (Cordts et al., 2016). The dataset contains 5000 high-resolution, densely annotated frames with 19 object classes.

Unlike in more general segmentation tasks, street view images are taken in environments that share many similarities between scenes and the distribution of object classes over the image plane is highly non-uniform. For instance, often large parts of the image are occupied by empty streets or the sky, while others have a higher density of different object classes. Thus, the segmentation task could benefit from distributing pixels to areas of higher information density.

As baselines, we choose PSPNet (Zhao et al., 2017), SegNeXt (Guo et al., 2022) and DeepLabV3+ (Chen et al., 2018). Again, we train the baseline with a fixed uniform pixel layout

| Resolution | Method | Accuracy | $\theta$ |
|---|---|---|---|
| $8 \times 8$ | ResNet18 | 86.60% | - |
| | ResNet18 + $\phi_{rect}$ | **87.19**% | (0.32, 0.08) |
| $16 \times 16$ | ResNet18 | 88.39% | - |
| | ResNet18 + $\phi_{rect}$ | **88.62**% | (0.15, 0.03) |

Table 2: Accuracy and pixel layouts for classification on the CelebA dataset (Liu et al., 2015).

and compare it to the same network architecture with a learnable layout. To assess if curvilinear or rectangular layouts have an advantage over the other, we train variants with both. We also evaluate if a layout optimized for one network can be transferred to another on the same task by transferring the optimal rectangular layout from PSPNet to the other networks. More experiments on different resolutions and per-class results can be found in Appendix C.

The learned layouts in Fig. 4b display a higher pixel density towards the horizon line. As expected, fewer pixels are needed to detect the street and the sky. Perspective also dictates that objects far away from the camera gather at the horizon and are smaller, thus requiring a finer pixel raster to distinguish them. Interestingly, the sensor with the rectangular layout also learned to put slightly more pixels towards the left and right edges. This might be due to the fact that there is often a higher density of small objects on the sidewalks (people, poles, fences, vegetation) as opposed to the street itself.The learned curvilinear layout does not show this horizontal non-uniformity. Although hard to verify, an explanation for this could be that the deformation here is restricted to a circular region, which means that the region at the sensor boundary can not be samples more densely.

The quantitative results in Tab. 1 show a clear advantage over the uniform layout on all tested resolutions. In all our experiments, the rectilinear layouts outperformed their curvilinear counterparts, which might be because of their limited adaptability in the image corners. The layouts transferred from PSPNet also achieve improved results, which indicates that the layouts are task specific rather than network specific. Fig. 5 also shows encouraging results for the detection of small objects in the image center. More learned layouts are visualized in Appendix D.

**Multi-label classification.** Finally, we evaluate our method on the multi-label classification of facial attributes on the CelebA dataset (Liu et al., 2015). Similarly to MNIST, the faces are in the image center, so that a non-standard pixel layout could be advantageous. We fine-tune a ResNet18 pretrained on ImageNet on different resolutions over 15 epochs with rectilinear pixel layout. The results in Tab. 2 show marginal improvements over the baselines. As expected, the network learns a higher density in the image center. We conjecture that the joint prediction of 40 different attributes (spatially spread over each face) is the reason for the comparably small improvements.

## 6 CONCLUSIONS

We present the first approach of a parameterizable and differentiable pixel layout that can be jointly optimized with any downstream, task-specific network in an end-to-end fashion. This approach allows going beyond fixed, regular pixel layouts that treat all regions of an image as equally important, yielding task specific layouts on which the networks perform better than on regular grids. We provide the generic concept of a differentiable sensor layout parameterization that retains a regular topology and present two pixel layout parameterizations, i.e. rectangular and curvilinear grid shapes. Our drop-in module allows applying the learnable pixel layout to existing high-resolution imagery, and, thus, connecting our method to existing deep-learning pipelines. We show that network predictions benefit from a learnable pixel layout for classification and segmentation tasks, and depict the CMOS hardware design process for a rectilinear pixel layout.

**Limitations.** The proposed deformation functions are comparably simple and do not cover all task preferences and require more investigation. In addition, not all tasks have a sufficient spatial bias, i.e., a non-uniform distribution of objects of interest in the image plane to make non-uniform sensor layouts advantageous. Moreover, a tighter integration of the sensor design parameters and their (geometric) limits into the modeling of deformations is needed.

## 7  ETHICS STATEMENT

We have carefully read the ICLR 2024 Code of Ethics and confirm that we adhere to it. The method we propose in this paper is conceived to jointly optimize deep neural network models for computer vision and the imaging sensor that records the input images, in order to benefit accuracy at lower overall sensor resolution. This can have positive effects on power consumption and compute time. However, since the proposed approach itself is fundamental research, it could in principle be used in diverse application scenarios. To exemplify a use-case that benefits society, we focus in this paper on semantic segmentation of street scenes, to potentially improve safety for example in driver assistance systems or autonomously driving cars.The reduced number of pixel used to acquire regions with high information density for the specific task, increases data privacy, as the acquisition of dispensable data is prevented.

## 8  REPRODUCIBILITY STATEMENT

We provide implementation details in Appendix A and Appendix B and code in the supplementary material. All code to reproduce results will be cleaned and made publicly available upon accep-tance.

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

## A    ADDITIONAL IMPLEMENTATION DETAILS

We implement our differentiable sensor as a custom PyTorch layer, which makes it easy to integrate into already existing projects. The layer takes a high-resolution image as input and outputs an image with the resolution of the simulated sensor. We use $\tanh$ to restrict the parameters to the allowed range of $(-1, 1)^2$. Since the behavior of the deformation function becomes nearly singular at the borders, we restrict the permissible range a bit more to $(-0.6, 0.6)^2$. This range already allows for strong enough deformations in our experiments. We will make our implementation of the differentiable sensor layer as well as code for reproducing the experiments shown in this paper publicly available upon acceptance.

Even though we simulate non-uniform layouts, we interpret the output of the sensor layer as a uniform grid, i.e., the output is just an image tensor in $\mathbb{R}^{3 \times R_1 \times R_2}$ without information about the size and relative distance of neighboring pixels. Directly visualizing this output makes the image look deformed (see Fig. 1). For instance, layouts with more pixels in the image center would result in enlarged objects in the image center. We feed this deformed image directly into the downstream network. Even though this seemingly violates the implicit assumption of convolutional layers that all pixels have the same distance to their neighbors, our experiments show that we still achieve significant improvements over baselines. Exploring more tailored solutions accounting for relative pixel positions, like continuous CNNs, should be addressed in future research.

Special care has to be taken if the downstream network is solving an image-to-image task. In that case, there is usually a one-to-one correspondence between input pixels and output pixels, like a dense map of class probabilities for semantic segmentation. This means that a change in the pixel layout of the input image has an immediate effect on the output image, which has to be taken into account when computing the loss. This will, in general, be some form of distance to a reference image given in uniform pixels. We therefore modify the loss function to properly account for pixel area. To this end, we warp a uniform grid with the reference image resolution $R_1^{\mathrm{gt}} \times R_2^{\mathrm{gt}}$ via the inverse pixel layout deformation $\phi^{-1}$ and sample the (deformed) output at these locations with nearest-neighbor interpolation. The result is an un-deformed image with constant regions that match the pixel layout defined by $\phi$. We can now take the loss to the reference image as usual. Note, that we only apply this resampling to the *output* of the downstream task while the network operates on the deformed image and can thus take advantage of the non-uniform layout. These considerations only apply to image-to-image tasks. If the output of a task does not depend on positions on the image plane (for instance classification tasks), the loss function can be applied as-is without modifications.

## B    DETAILS ON ARCHITECTURES AND TRAINING PROCEDURES

In order to achieve the fairest possible comparisons, the uniform baselines and the learned layouts share the *exact same* code (including training schemes and hyperparameters), except that the baselines have the sensor layout parameters frozen to 0, which is equivalent to a uniform layout. In the following sections we will give more details for each experiment. We have no affiliations with the authors of any mentioned Github repositories. All code will be made publically available upon acceptance.

**MNIST classification experiments**    The sensor layer output is fed into a small CNN consisting of two convolutional layers with 32 and 64 channels, followed by a max pooling layer and two fully connected layers with hidden dimension 128. We train the whole pipeline for 14 epochs with Adam (Kingma & Ba, 2014) and a learning rate of 0.01. As a baseline, we compare against exactly the same pipeline and training scheme, but with frozen uniform sensor parameters.

**Semantic segmentation experiments**    PSPNet (Zhao et al., 2017) and PSANet (Zhao et al., 2018) implementations are taken from https://github.com/hszhao/semseg and the training follows the default configurations for cityscapes. For both we use a ResNet50 backbone. In general, segmentation networks are trained and evaluated on patches of the high-resolution input images, instead of downsampling them. Since our objective is to optimize the pixel layout over the whole sensor plane, we instead feed the whole input to the network directly. This also means that we cannot employ all data augmentation schemes from Zhao et al. (2017), namely random rotation, which we disabled. This specific implementation requires all (network input resolutions + 1) to be divisible

by 8, so we use resolutions of e.g. $257 \times 129$ instead of $256 \times 128$. This difference of 1 pixel is negligible. We initialize all sensor parameters with 0 (uniform layout) and freeze them for the first 10 epochs to speed up convergence to reasonable network weights. In total we train each networks for 200 epochs.

Implementations for SegNext (Guo et al., 2022) and DeepLabV3+ (Chen et al., 2018) are taken from https://github.com/open-mmlab/mmsegmentation. Again, we use the default configurations for Cityscapes. Namely, we use the config files segnext_mscan−t_1xb16−adamw−160k_cityscapes−512x512.py and deeplabv3plus_r18b−d8_4xb2−80k_cityscapes−512x1024.py as a base. Like for PSPNet we made modifications to take the whole image as network inputs instead of working on patches and removed data augmentation schemes that change the camera orientation. Apart from that, no training scheme or network architecture changes were made.

**CelebA multi-class classification**   We fine tune a ResNet18 pretrained on ImageNet provided by torchvision for 15 epochs, a batch size of 64 with an Adam optimizer and a learning rate of 0.001.

## C   ADDITIONAL SEGMENTATION RESULTS

**Per-class metrics**   We give per-class IoU results for the cityscapes testset in Table 4. We provide additional experiments using PSANet (Zhao et al., 2018) as a backbone. Using a learned sensor layout consistently leads to better performance compared to training the network with a uniform layout.

**Oracle performance**   As explained in the paper, we compare the predicted segmentation masks with the ground truth segmentation by resampling the (non)-uniform network outputs to the higher resolution GT masks. Thus, even perfect segmentation masks have an inherent error. An interesting question is how this inherent error differs between the standard uniform layout and our learned layouts.

To this end, we compute the theoretically best achievable error for each layout with the following scheme. First, we encode the ground truth segmentation masks with one-hot vectors at each pixel to get tensors of shape $19 \times W \times H$, i.e. a tensor of class probabilities. Then we use our sensor layout to downsample these tensors with the different sensor layouts. We convert the result back to a segmentation mask by taking the channel index with the highest probability. Finally, this mask is upsampled to the reference image resolution and evaluated as usual.

The result can be found in Tab. 3. While there seems to be an inherent advantage of the learned layouts at lower resolutions, the highest attainable performance at $512 \times 256$ becomes worse. Thus, we hypothesize that the increased performance of our method in the paper mostly stems from the deep learning model taking advantage of the increased resolution in areas of the sensor that are more challenging.

**Influence of pixel density on accuracy**   We compare the accuracy gain of our learned layouts compared to the uniform baseline by computing the mean accuracy at every pixel positions. Fig. **??** exemplary shows the accuracy gain of the learned layouts with the PSPNet backbone. It can be seen that the accuracy significantly increases in regions with higher pixel density, while only staying the same or only slightly decreasing in low density regions. This is evidence that the learned layout successfully puts higher density into regions that are otherwise hard for the network to discriminate while putting less emphasis into easier areas (like the street).

Furthermore, the area of accuracy increase over the uniform layout seems to broaden in vertical direction towards the left and right corners for the rectangular layout. This is an area that can not easily be attended to by the curvilinear layout parameterization without high distortions.

## D   LEARNED LAYOUTS

We give visualizations of all learned layouts in the paper, namely for all segmentation networks in Fig. 7 and classification networks in Fig. 10.

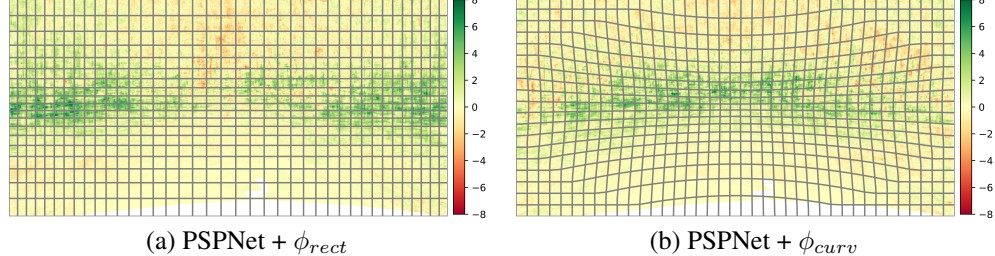

(a) PSPNet + $\phi_{rect}$        (b) PSPNet + $\phi_{curv}$

Figure 6: Difference of per pixel accuracy in percentage points compared to the uniform layout for $256 \times 128$ PSPNet on Cityscapes. Values greater than 0 indicate image regions where the learned layout performs better. The per pixel accuracies were computed as the average accuracy over the whole test set. The corresponding layouts are overlaid and do not represent the actual resolution of $256 \times 128$ for visualization purposes.

| Resolution | Layout | mIoU (%) | Acc. (%) |
|---|---|---|---|
| $128 \times 64$ | Uniform | 84.29 | 96.06 |
| | $\phi_{curv}$ | 84.69 | 96.13 |
| | $\phi_{rect}$ | **85.02** | **96.16** |
| $256 \times 128$ | Uniform | 90.71 | 97.57 |
| | $\phi_{curv}$ | **90.96** | 97.53 |
| | $\phi_{rect}$ | 90.93 | **97.60** |
| $512 \times 256$ | Uniform | **94.88** | **98.77** |
| | $\phi_{curv}$ | 94.83 | 98.73 |
| | $\phi_{rect}$ | 94.39 | 98.66 |

Table 3: Theoretical best achievable performance with our learned layouts compared to uniform layout. The learned layouts are the layouts we trained together with PSPNet on the cityscapes dataset from our other experiments. While there is an *inherent* but small advantage of our learned layout on lower resolutions, this advantage diminishes at the highest resolution.

| Resolution | Method | road | swalk | build. | wall | fence | pole | tlight | sign | veg. | terrain | sky | person | rider | car | truck | bus | train | mbike | bike | mIoU | mIoU/GFLOP |
|---|---|---|---|---|---|---|---|---|---|---|---|---|---|---|---|---|---|---|---|---|---|---|
| $128 \times 64$ | PSPNet | 93.8 | 59.0 | 77.5 | 24.3 | 11.8 | **7.8** | 6.2 | 14.3 | 77.6 | 43.8 | **80.6** | 33.6 | 6.0 | 75.5 | 29.4 | 30.6 | 10.2 | 0.2 | 22.9 | 37.1 | 6.6 |
| | + $\phi_{curv}$ | 94.2 | 60.9 | 77.4 | 23.4 | 11.5 | 7.1 | **6.6** | 13.5 | 77.7 | 46.2 | 80.3 | 33.1 | 8.2 | 76.3 | 20.2 | 31.2 | **11.8** | **10.2** | 23.8 | 37.6 | 6.7 |
| | + $\phi_{rect}$ | **94.5** | **62.5** | **78.3** | **28.5** | **16.7** | 7.3 | 5.2 | **15.5** | **78.2** | **48.4** | 80.2 | **35.7** | **8.3** | **78.4** | **39.0** | **35.3** | 3.1 | 5.1 | **29.2** | **39.4** | 7.0 |
| $256 \times 128$ | PSANet | 95.8 | 69.0 | 83.5 | 32.6 | 27.7 | 18.7 | 22.4 | 36.5 | 84.0 | 54.2 | 87.4 | 50.0 | 22.8 | 84.8 | 43.0 | 47.9 | 32.7 | 19.3 | 46.6 | 50.4 | 2.0 |
| | + $\phi_{curv}$ | 96.1 | 70.8 | 83.9 | 35.2 | 31.0 | 17.4 | 23.3 | 37.4 | 84.1 | 54.8 | 87.2 | 50.2 | 27.0 | 86.1 | 49.9 | 50.4 | 22.1 | 29.7 | 47.8 | 51.8 | 2.1 |
| | + $\phi_{rect}$ | 96.4 | 73.1 | 84.0 | 38.1 | 33.2 | 19.5 | 24.6 | 37.4 | 84.4 | **58.1** | 86.6 | 52.2 | 29.0 | 86.3 | 52.5 | 52.9 | 26.0 | 26.3 | **51.4** | 53.3 | 2.1 |
| | PSPNet | 96.0 | 70.1 | 83.5 | 35.7 | 27.3 | 18.4 | 23.1 | 37.2 | 83.8 | 54.2 | 87.1 | 49.8 | 23.4 | 84.8 | 39.6 | 49.8 | 35.2 | 19.2 | 46.3 | 50.7 | 2.2 |
| | + $\phi_{curv}$ | 96.3 | 71.9 | 83.7 | 39.9 | 29.3 | 17.5 | 22.2 | 36.7 | 84.0 | 55.8 | 86.6 | 51.3 | 28.7 | 86.2 | **59.1** | 51.2 | 20.6 | 25.4 | 49.0 | 52.4 | 2.3 |
| | + $\phi_{rect}$ | **96.6** | 73.2 | 84.1 | 39.2 | 34.0 | 19.6 | 24.1 | 37.9 | 84.2 | 57.2 | 86.5 | 52.6 | 30.7 | 86.4 | **59.1** | 54.2 | 34.7 | 23.0 | 50.4 | 54.2 | 2.4 |
| | SegNeXt | 95.3 | 66.5 | 82.5 | 37.4 | 26.3 | 15.6 | 23.7 | 33.2 | 82.8 | 51.4 | 86.9 | 46.3 | 20.3 | 83.1 | 49.2 | 43.6 | 24.2 | 20.5 | 45.4 | 49.2 | 12.0 |
| | + $\phi_{rect}$ | 95.4 | 66.7 | 82.5 | 33.0 | 28.6 | 15.4 | 24.7 | 33.5 | 83.0 | 54.0 | 87.0 | 47.2 | 20.1 | 83.4 | 46.7 | 46.4 | 31.1 | 25.7 | 45.9 | 50.0 | 12.2 |
| | +$^{\dagger}\phi_{rect}$ | 95.8 | 69.4 | 82.7 | 42.1 | 31.6 | 16.7 | 22.7 | 31.0 | 82.8 | 53.0 | 84.8 | 47.6 | 23.9 | 84.5 | 53.2 | 48.7 | 27.7 | 23.8 | 48.5 | 51.1 | 12.5 |
| | DeepLabv3+ | 96.5 | 71.8 | **84.6** | **42.5** | 32.1 | 27.7 | **27.4** | **40.7** | 84.6 | 51.6 | **89.8** | 53.5 | 28.6 | 86.5 | 45.7 | 61.2 | 45.9 | 30.0 | 46.8 | 55.1 | 2.5 |
| | + $\phi_{rect}$ | 96.1 | 73.3 | **84.6** | 42.3 | 32.3 | 27.2 | 26.4 | 40.4 | **84.9** | 55.1 | 89.2 | **54.9** | **32.7** | **87.5** | 56.7 | 63.4 | **49.7** | **30.2** | 49.7 | **56.7** | 2.6 |
| | +$^{\dagger}\phi_{rect}$ | 96.7 | **73.9** | 84.5 | 44.2 | **34.4** | **28.5** | 24.8 | 39.5 | 84.5 | 53.5 | 88.5 | 54.4 | 30.6 | 87.1 | 57.1 | **63.5** | 36.8 | 29.3 | 51.3 | 56.0 | 2.5 |
| $512 \times 256$ | PSPNet | 97.2 | 77.9 | **88.4** | 48.7 | 44.7 | **37.3** | **47.3** | **59.6** | **88.7** | 58.8 | **91.7** | **65.6** | 42.0 | 90.4 | 56.0 | 74.0 | 48.2 | 36.2 | 62.8 | 64.0 | 0.71 |
| | + $\phi_{curv}$ | **97.3** | 78.8 | 88.2 | 48.2 | 45.7 | 35.7 | 45.1 | 57.3 | 88.6 | 60.0 | 91.1 | 65.2 | 43.1 | **90.7** | 60.0 | **74.6** | **51.5** | **43.6** | **63.8** | **64.7** | 0.72 |
| | + $\phi_{rect}$ | **97.3** | **79.4** | 88.2 | **51.8** | **46.6** | 36.4 | 44.2 | 56.0 | 88.3 | **62.4** | 90.3 | 64.2 | **43.9** | 90.6 | **65.9** | 71.8 | 49.2 | 40.0 | 63.7 | **64.7** | 0.72 |

Table 4: Per-class results (IoU). For lower resolutions, the proposed approach allows improvements over regular layouts for most classes, yet in particular for important classes of small objects such as *person* (by almost 3%) or *rider* (up to 7%).

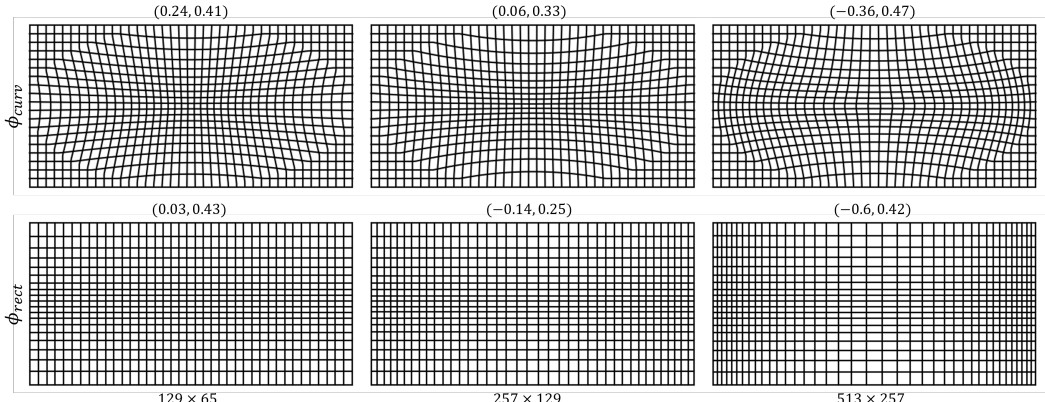

Figure 7: Learned layouts for all segmentation experiments with PSPNet. The parameters are on top of each layout. The resolution of the layouts is lowered for visualization purposes.

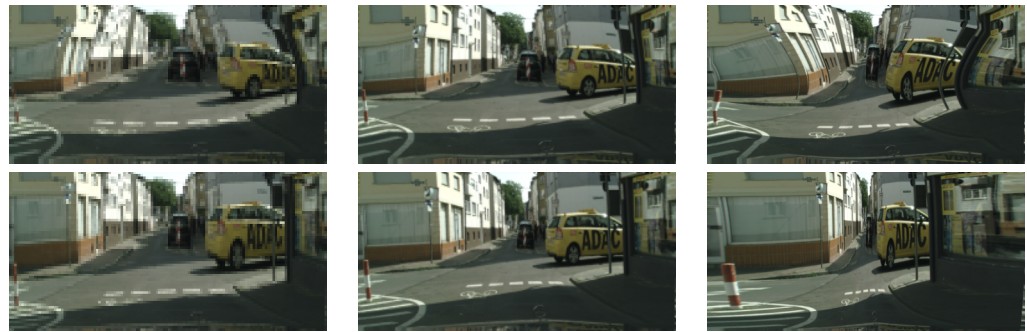

Figure 8: The corresponding sensor images to the layouts in Fig. 7, interpreted as being on a uniform layout. This means regions with high pixel density are enlarged.

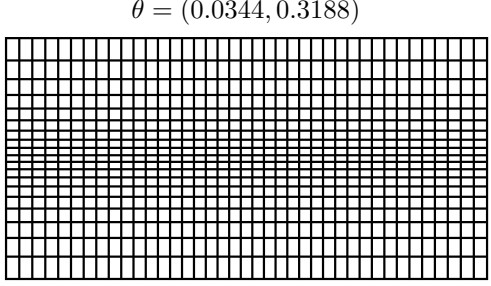

Figure 9: Pixel Layout for DeepLabv3+ with Layout learned from scratch.

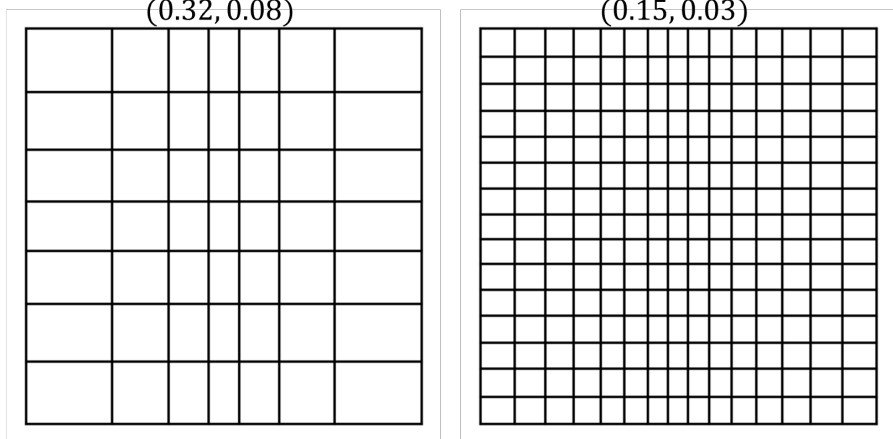

Figure 10: Learned layouts for the classification experiments on CelebA. Both layouts are at the actual resolutions of $8 \times 8$ and $16 \times 16$

## E  IC LAYOUTS

We designed and layed out the image sensor using $0.18\mu m$ XFAB CMOS Image Sensor process. Typical 4-transistor active pixel sensors used in most digital cameras were used. Transistors were selected to be minimum size, which in this process is $0.35\mu m \times 0.22\mu m$. The transfer gate required for correlated double sampling for fixed pattern and temporal noise removal is $0.8\mu m \times 1\mu m$, due to minimum size requirements imposed by the foundry. We designed a standard pixel of $5\mu m \times 5\mu m$, which would have been a typical image sensors pixel. It is comparatively larger than the state of the art; however, was designed to showcase the concept and hence this can be scaled down. In Fig. 11, the red areas denote the photodiode, while the rest covered with the third and top metal layer are the 4 transistors used to reset and read out the pixel. Power is provided using the third metal in blue, while the top metal is only used for optical shielding. The output line is the long running second metal, in yellow; while the reset and transfer gate are in first metal in white; while the row select is in the third metal in blue. These base pixels were then placed at the starting point of each coordinate, as predicted from analytical calculations and rounded off to the nearest $0.18\mu m$, to ensure manufacturability. The space between neighboring pixels was then filled with photodiodes of different dimensions and merged with the photodiode of the base pixel, to lead to pixels of different size. Despite starting with a large pixels, our entire array size is $2mm \times 1.4mm$, which is a fairly small imaging chip.Furthermore, despite quantisation in pixel boundaries, each pixel is still fairly individual compared to its nearest neighbors. The smallest pixel in the array is $6.534\mu m \times 5.76\mu m$, showing a small addition to photodiode to the base pixel. The largest pixel is $17.91\mu m \times 10.26\mu m$, showing a six times larger area than the base pixel. This would lead to a very small reduction in the overall intensity dynamic range of the pixel; however, this should not affect typical natural world scenes. If it does, one can use a modified dual sampling technique, wherein the larger pixels would be read with a shorter integration time and the smaller pixels read with longer integration time. This is feasible in the current array due to its random addressability. Furthermore, the row scanners, column scanners, column readout circuits and the analogue to digital converters present in each column is designed to match the pixel pitch of the base pixel. Hence, whenever these are controlling larger pixels, space is left between neighboring row or column circuit to compensate for pixel sizes while continuing the signal chain.

The layouts for different regions of the sensor (and thus different pixel shapes) can be seen in Fig. 12.

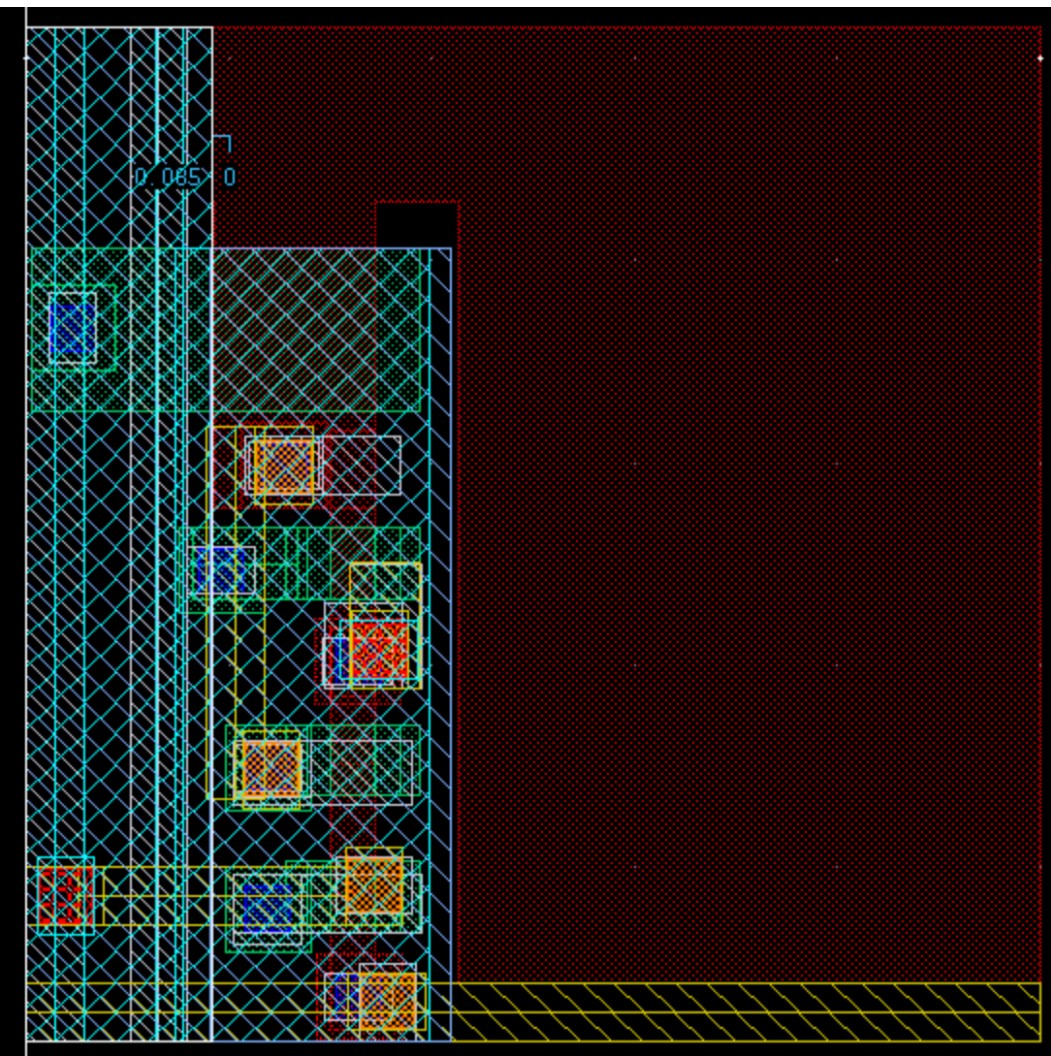

Figure 11: Base Pixel

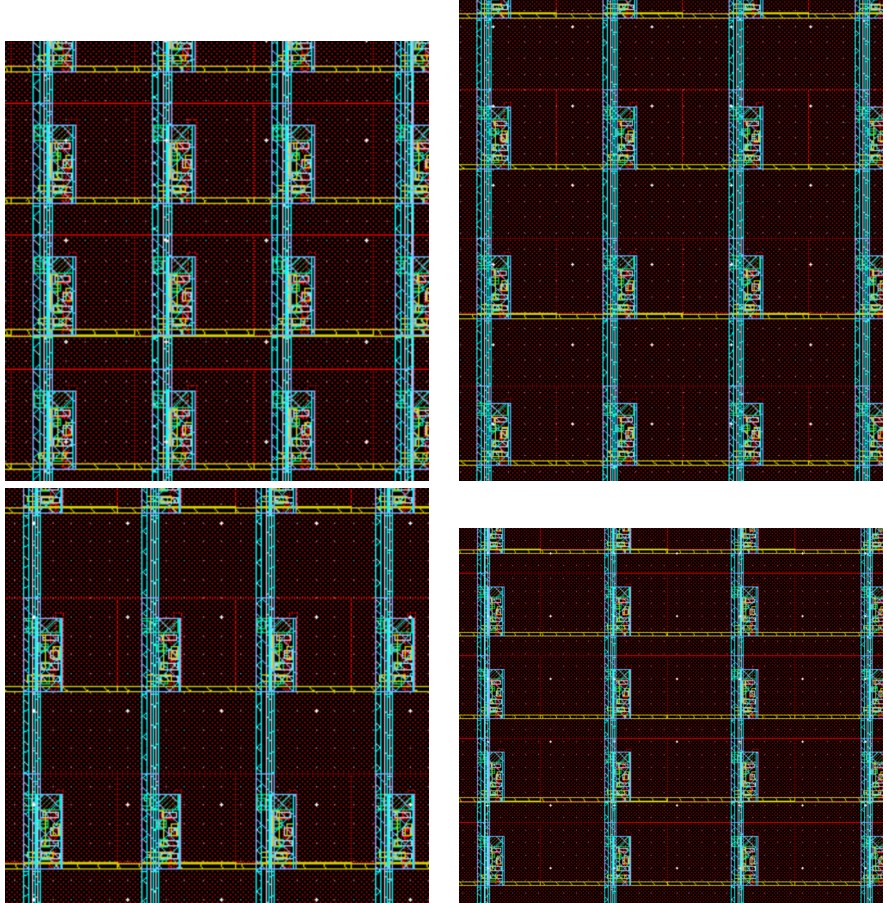

Figure 12: IC layouts in different regions of the optimized sensor for the $256 \times 128$ Cityscapes segmentation experiment. The base pixel (square region in the bottom left of each pixel) is always the same size.

