# OpenReview forum: "Differentiable Sensor Layouts for End-to-End Learning of Task-Specific Camera Parameters"
_ICLR.cc/2024/Conference — Submitted to ICLR 2024_

### Official Review · Reviewer_TZ9J · 2023-10-31

**Soundness:** 3 good
**Presentation:** 3 good
**Contribution:** 3 good
**Rating:** 8
**Confidence:** 3

**Summary:**

This paper presented advances in integrating AI and hardware sensing design to more cost and energy-efficient solutions by optimising hardware parameters for task-specific problems in an end-to-end manner. The central proposition is in learning task-specific pixel layout parameterisation. To this end, this paper proposes a sensor simulation framework that allows end-to-end training, and  a pixel layout parameterisation. Initial experimentation confirms performance benefits on learned layouts over classification, semantic segmentation and multi-label classification.

**Strengths:**

The paper is well presented and motivated. Focusing on energy and resource-efficient solutions is attractive and, in my opinion, an important research direction. Concepts were clearly explained at the correct level of detail to transmit key ideas and propositions. Experimentation, although limited, confirmed intuition and the capacity to learn pixel layouts end-to-end.

**Weaknesses:**

My main criticism is in experimentation, which could be more extensive in the number of datasets and problem configurations. For example, assessing the performance gain from learning layout parameterisation across a range of image resolutions could provide more insight into the applicability of this research.

Fig. 1 caption should be more descriptive of the proposed pipeline.

**Questions:**

Do authors know the performance gain over very low resolutions for image segmentation?

---

> ### Author Response · Authors · 2023-11-15
>
> Thank you for your positive feedback and the suggestions to improve our paper!
>
> >My main criticism is in experimentation, which could be more extensive in the number of datasets and problem configurations. For example, assessing the performance gain from learning layout parameterisation across a range of image resolutions could provide more insight into the applicability of this research.
>
> We evaluated the PSPNet baseline with both the curvilinear and rectangular layouts on additional resolutions:
>
> | Resolution | Method          | mIoU |
> |------------|-----------------|------|
> | 128 x 64   | PSPNet          | 37.1 |
> | 128 x 64   | + $\phi_{curv}$ | 37.6 |
> | 128 x 64   | + $\phi_{rect}$ | **39.4** |
> | 256 x 128  | PSPNet          | 50.7 |
> | 256 x 128  | + $\phi_{curv}$ | 52.4 |
> | 256 x 128  | + $\phi_{rect}$ | **54.2** |
> | 512 x 256  | PSPNet          | 64.0 |
> | 512 x 256  | + $\phi_{curv}$ | **64.7** |
> | 512 x 256  | + $\phi_{rect}$ | **64.7** |
>
> These results as well as per-class metrics can be found in Tab. 4 in the appendix. We also added visualizations of all learned layouts of the above experiments and example sensor output images in Fig. 7 and 8.
>
> >Do authors know the performance gain over very low resolutions for image segmentation?
>
> The lowest resolution we tested was 128x64 with PSPNet on cityscapes, which is a factor 16 lower than the original cityscapes resolution of 2048x1024. Our learned rectangular layout achieved an mIoU of 39.4 while the uniform layout achieved an mIoU of 37.1.
>
> We are happy to answer anything that is still unclear or any further questions you have.

---

### Official Review · Reviewer_ZrRZ · 2023-11-01

**Soundness:** 3 good
**Presentation:** 3 good
**Contribution:** 2 fair
**Rating:** 5
**Confidence:** 4

**Summary:**

- The authors present a method to optimize pixel layout on an imaging sensor for a specific task.
- To represent differentiable sensor layout, two pixel parameterization functions are proposed: rectangular and curvilinear.
- A drop-in module that approximates sensor simulation given existing high-resolution images can be easily incorporated into existing deep learning models.
- The authors show that task like semantic segmentation in autonomous driving can benefit from non-uniform pixel layouts.

**Strengths:**

- The differentiable sensor layout parameterization allows for task-specific, local varying pixel resolutions, which can improve the performance of deep learning models for tasks like semantic segmentation in autonomous driving.
- The authors define a class of pixel layouts to be a parameterized deformation function which is required to be bijective and bi-Lipschitz, implying that the function is differentiable to enable end-to-end training.

**Weaknesses:**

- [Generalization in diverse applications] The experiments in this paper are limited to specific tasks such as semantic segmentation and multi-label classification on facial attributes, so it is unclear how well the proposed method would generalize to other computer vision tasks.
- [Generalization in different scenes] The authors propose a simple deformation, so additional experiments are required to see if it is effective in datasets with anomalous scene or in robotics tasks with simultaneous indoor and outdoor scenes. How effective is it in covering a variety of scene structures with only two parameters (theta_1, theta_2)?
- [Exp. on computational cost] In Sec. 2 (in paragraph [End-to-end Optimization of the ISP pipeline]), the authors mention that the proposed model can reduce the size of the network and the training time, so further experiments on the computational cost of the proposed method are needed.
- [Comparison with non-uniform] The authors conduct experiments comparing their method to other method (Zhao et al., 2017) using a uniform layout, and additional experiments comparing their method to other method using non-uniform layout are needed. (Marin et al., 2019)
- [Exp. on different object size] In Sec. 5 (in paragraph [Semantic Segmentation]), the authors argue that rectangular layout is learned to put more pixels towards the left and right edges because of a higher density of small objects on the sidewalks and to confirm this effect, experimental results based on the class of small objects near the horizon or accuracy in dense area is required to support this effect.
- [Exp. on different resolution] The authors run all of their experiments at a lower resolution, but a comparison with experiments at the original resolution along with the computational cost is needed as well.

**Questions:**

- The authors say that the rectilinear layouts outperformed curvilinear layouts in all experiments because of curvilinear layouts’ limited adaptability in the image corners, more detailed explanation of this part is needed.
- In Sec. 5 (in paragragh [Semantic Segmentation]), are there any experiments on accuracy by dense area or class to demonstrate the effectiveness of the learned pixel layout? (as commented in 6. Weakness [Exp. on different object size])

---

> ### Author Response · Authors · 2023-11-15
>
> Thank you for your thoughtful and detailed review! We will respond to each of your comment one by one.
>
> >[Generalization in diverse applications] The experiments in this paper are limited to specific tasks such as semantic segmentation and multi-label classification on facial attributes, so it is unclear how well the proposed method would generalize to other computer vision tasks.
>
> Our method aims to exploit spatial bias in the input data distribution to enhance a specific computer vision task and as such the learned sensor layouts are highly tuned to that data/task combination. This is especially useful for tasks like semantic segmentation for autonomous driving, because the scene structure has that clear spatial bias.
> Common datasets and benchmarks for more low-level computer vision tasks like denoising or deblurring do not have a sufficient spatial bias, presumably since the images are taken from a very broad distribution. Nevertheless, if a spatial bias is present, we expect our method to be applicable to larger computer vision pipelines. However, there is no simple way to assess the amount of spatial bias.
>
> >[Generalization in different scenes] The authors propose a simple deformation, so additional experiments are required to see if it is effective in datasets with anomalous scene or in robotics tasks with simultaneous indoor and outdoor scenes. How effective is it in covering a variety of scene structures with only two parameters (theta_1, theta_2)?
>
> We agree that our deformation model is comparatively simple, but want to underline that even such a simple model has significant improvements, as shown in the paper. While more expressive deformation models are generally possible (such as more general diffeomorphisms based on normalizing flow), it is hard to incorporate the specific constraints, i.e., computational efficiency, boundary consistency, and manufacturability in hardware.
>
> We'd like to point out that the concept of an optimized sensor layout is *task-specific* and applicable to any task with an inherent spatial bias. The more universal the requirements get (e.g. working in significantly different indoor and outdoor scenes) the less we expect to benefit from a task-specific optimization. The particular case of a moving robot could, of course, make a spatially varying resolution very interesting when adapting the robots' movements in order to focus high resolution parts on the object of interest (similar to how the human visual system works). We consider this to be an extremely interesting and promising direction of future research, but - as no prior work has even considered the optimization of the sensor layout - to be beyond the scope of this work, where the general methodology for sensor layout optimization needed to be derived first.
>
> >[Exp. on computational cost] In Sec. 2 (in paragraph [End-to-end Optimization of the ISP pipeline]), the authors mention that the proposed model can reduce the size of the network and the training time, so further experiments on the computational cost of the proposed method are needed.
>
> This is a typo, thank you for pointing it out. Our method reduces inference time, because once the sensor is built, our method does have no computational overhead while allowing the network to operate on a lower resolution. As the FLOPS scale roughly linearly with the number of pixels, lower resolutions can have a more favorable accuracy / FLOPS tradeoff than higher resolutions. We added this tradeoff to Tab. 4 in the appendix.
>
> >[Comparison with non-uniform] The authors conduct experiments comparing their method to other method (Zhao et al., 2017) using a uniform layout, and additional experiments comparing their method to other method using non-uniform layout are needed. (Marin et al., 2019)
>
> The comparison to (Marin et al. 2019) would be unfair, since we try to solve a different problem. (Marin et al. 2019) compute non-uniform downsampling patterns on a per-image basis. They require a high resolution image (and thus also a high resolution sensor) to compute these patterns. On the other hand our approach aims to optimize a low resolution (but fixed) sensor layout in a data driven fashion by exploiting the inherent spatial biases in a dataset.
> Our method only needs high resolution images for the differentiable sensor simulation during training. During inference (with a built sensor) we do not need high resolution images at all anymore and can directly feed the sensor output to the downstream network - without any computational overhead.

---

> ### Author Response · Authors · 2023-11-15
>
> >[Exp. on different object size] In Sec. 5 (in paragraph [Semantic Segmentation]), the authors argue that rectangular layout is learned to put more pixels towards the left and right edges because of a higher density of small objects on the sidewalks and to confirm this effect, experimental results based on the class of small objects near the horizon or accuracy in dense area is required to support this effect.
>
> We now compute average per-pixel accuracies over the test set which we visualize in Fig. 6 in Appendix C. The visualizations show that performance indeed increases in regions of higher densities compared to the uniform layout, which in the case of the rectangular layout lies at the horizon line and towards the left and right image edges.
>
> >[Exp. on different resolution] The authors run all of their experiments at a lower resolution, but a comparison with experiments at the original resolution along with the computational cost is needed as well.
>
> An accurate sensor simulation needs multiple samples per pixel, which cannot be achieved at the original resolution without additional information.
>
> >The authors say that the rectilinear layouts outperformed curvilinear layouts in all experiments because of curvilinear layouts’ limited adaptability in the image corners, more detailed explanation of this part is needed.
>
> The worse performance due to limited adaptability of the curvilinear layout is a hypothesis based on the numerical results we have gotten. Yet, this hypothesis is difficult to verify, particularly because the simulation is based on a high resolution rectangular grid, which could introduce a natural bias to rectangular pixels. We made this aspect more clear in the revised version of our paper in the experimental section and also in the very last section of Appendix C.
>
> >In Sec. 5 (in paragraph [Semantic Segmentation]), are there any experiments on accuracy by dense area or class to demonstrate the effectiveness of the learned pixel layout? (as commented in 6. Weakness [Exp. on different object size])
>
> Please refer to our comment above. We added visualizations of per-pixel accuracies in Fig. 6 in Appendix C.
>
> We are happy to address any further questions or parts that remained unclear.

---

### Official Review · Reviewer_9y41 · 2023-11-01

**Soundness:** 3 good
**Presentation:** 3 good
**Contribution:** 3 good
**Rating:** 8
**Confidence:** 5

**Summary:**

The paper proposes a differentiable sensor layout optimization approach for end-to-end task specific optimization. Conventional camera design optimizes different components such as, sensor, optics, ISP independently and there has been a recent push in making each of these stages to perform end-to-end differentiable task-specific optimization. There has been prior work on optics and ISP optimization but nothing on sensor layout optimization. This paper proposes to optimize the sensor layout using two pixel layout parameterization. The paper shows sensor layout optimization for a classification and a semantic scene segmentation task and shows improvement for the learned layouts.

**Strengths:**

There isn't any work on sensor layout optimization so it's a novel contribution in terms of task-specific layout optimization. Furthermore, the work shows a realization of the learned layout showing the manufacturability of the approach.

The paper shows experiments using different tasks and networks to compare the performance of learned layout.

**Weaknesses:**

The paper does not provide any details on how the manufactured layout was tested with real data.

Typically optics is optimized for the pixel pitch which would be difficult for non-homogeneous layout and increases the complexity of the optics. However, this can be mitigated using task-specific learned optics.

The paper ignores CFA in the optimization process which can have an effect on the color of the image resulting in negative impact on the certain color-dependent tasks.

**Questions:**

How was the manufactured sensor tested with real captures?

---

> ### Author Response · Authors · 2023-11-15
>
> Thank you for your positive assessment of our work and the insightful comments! We will respond to each of your questions and comments individually.
>
> >Typically optics is optimized for the pixel pitch which would be difficult for non-homogeneous layout and increases the complexity of the optics. However, this can be mitigated using task-specific learned optics.
>
> Accurate simulation of optics was outside of the scope of our work, however we also assume that our sensor simulation layer could be fairly easily incorporated as a plug-in module in existing differential optics optimization methods.
>
> >The paper ignores CFA in the optimization process which can have an effect on the color of the image resulting in negative impact on the certain color-dependent tasks.
>
> We agree and believe that CFAs (maybe with filters that are learned in tandem with the pixel layout) could be an interesting avenue for future work.
>
> >How was the manufactured sensor tested with real captures?
>
> Once manufactured, the sensor is fixed and can not be changed at inference time. It is to be expected that it performs well as long as the data seen at inference time has the same inherent prior as during training. For the application-case of street scene recordings from a driving car, extensive validation on real captures would require close collaboration with the automotive industry, such that we have to leave this aspect for future work. In particular, so far all reported experimental results rely on
> simulated data.
>
> In the paper we sketched the general approach to design non-uniform layouts and provide IC layouts, which has been formally verified to confirm the manufacturability of non-uniform pixel shapes. The chip is in the process of being submitted for fabrication using XFAB’s XS018 and will be manufactured in March 2024, for which we are currently preparing the technical preliminaries.
> After the finalization of the design, its layout, interface circuits and tape-out, it typically requires further 6 months until the chip is fabricated. After fabrication of the chip, the testing of the chip's functionality and the integration into a camera takes another 6 months, so we expect to have an operative prototype camera in about 12-18 months from now.
>
> We apologize in case there was a misunderstanding and have clarified the paper in this respect by adding a paragraph at the end of Chapter 5 and a footnote in the introduction.

---

### Official Review · Reviewer_iE4U · 2023-11-02

**Soundness:** 2 fair
**Presentation:** 2 fair
**Contribution:** 3 good
**Rating:** 5
**Confidence:** 4

**Summary:**

This work presents a differentiable trained imaging pipeline to optimize sensor parameters and network parameters. The author presents a differentiable sensor simulation framework that can be easily integrated with the current neural network optimization framework to jointly optimize the sensor configurations and network parameters.

**Strengths:**

* The proposed optimization framework is fully differentiable in a physically plausible manner.
* The framework is flexible and can adapt to different types of camera-based tasks.

**Weaknesses:**

* How are the Deeplabv3+, PSPNet, SegNetXt trained? It seems the reported performance of Deeplabv3+ on the original image is lower than the original paper. Is this the reproduced result following official GitHub repo parameters? If the original model is not trained properly, then it is hard to distinguish if the performance boost is from extra fine-tuning (more training epochs) or the change of sensor parameters. I would encourage the author to provide more details.
* Is the designed hardware sensor in Sec. 4 evaluated in simulation or the real world?
* More visualizations for sensor images in cityscapes with learned layouts are encouraged as this can better help the reader understand how this sensor can influence the visual output. The data in MNist has very low image resolution thus the sampled visual output is too vague. More visualizations for high-resolution images are needed.
* What is the inference/training speed advantage of using the proposed method? The author claimed the speed advantage in the introduction. More quantitative results are needed to justify this argument.

**Questions:**

* What is the meaning of the red and green arrows in Figure 1? Some captions would help.
* Will the framework change the camera parameters for each sample? Or the parameters are learned from a dataset and once learned, it is fixed for evaluation and inference? The training pipeline for the whole system is still vague. A high-level description of the general framework would be also needed. Please specify.

---

> ### Author Response · Authors · 2023-11-15
>
> Thank you for the detailed questions on our paper presentation. We address them in the following one by one:
>
> >How are the Deeplabv3+, PSPNet, SegNetXt trained? It seems the reported performance of Deeplabv3+ on the original image is lower than the original paper. Is this the reproduced result following official GitHub repo parameters? If the original model is not trained properly, then it is hard to distinguish if the performance boost is from extra fine-tuning (more training epochs) or the change of sensor parameters. I would encourage the author to provide more details.
>
> We give details on the training procedure for all baselines in Appendix B. Specifically, we choose the same (default) hyperparameters for the training process of both the uniform baselines and their corresponding experiments with a learned sensor layout.
>
> The network and evaluation in the original Deeplabv3+ paper operates on the full resolution of cityscapes, which is 2048x1028, while the inputs to the networks in our method by design have much lower resolutions (i.e. 256x128).
>
>
> >Is the designed hardware sensor in Sec. 4 evaluated in simulation or the real world?
>
> All evaluations in the paper are made on simulated data. We make this clearer in the beginning of Chapter 5 in the revised manuscript.
>
>
> >More visualizations for sensor images in cityscapes with learned layouts are encouraged as this can better help the reader understand how this sensor can influence the visual output. The data in MNist has very low image resolution thus the sampled visual output is too vague. More visualizations for high-resolution images are needed.
>
> We add more outputs of the sensor layer in Fig. 8 in the Appendix. Since it is hard for the human eye to see differences between pixel sizes on high resolutions, we show images before resampling, which is the same images the networks get as inputs. This showcases which regions are enlarged by the sensor.
>
>
> >What is the inference/training speed advantage of using the proposed method? The author claimed the speed advantage in the introduction. More quantitative results are needed to justify this argument.
>
> The speed advantage mainly comes from the lower resolution. The sensor simulation is only needed during training. During inference (with a built sensor) the sensor image can be sent directly to the downstream network with no computational overhead.
> Concurrent methods like (Marin et al. 2019) and (Jin et al. 2021) downsample on a per-image basis, thus still requiring high resolution sensors (which require more energy) and introducing runtime overhead (computing the sampling positions + the downsampling operation itself).
>
>
> >What is the meaning of the red and green arrows in Figure 1? Some captions would help.
>
> Thank you for the suggestion. The green arrows indicate the data flow from the simulation input and the sensor simulation parameters through the whole end-to-end pipeline while the red arrows indicate the gradient flow back to the parameters in the backward pass. We specified the meaning of the arrow colors by adding a legend to Fig. 1.
>
>
> >Will the framework change the camera parameters for each sample? Or the parameters are learned from a dataset and once learned, it is fixed for evaluation and inference?
>
> The latter: The camera parameters are learned from data and fixed at inference time. We emphasize this aspect in our revised paper in the first section of Chapter 5.
>
>
> >The training pipeline for the whole system is still vague. A high-level description of the general framework would be also needed. Please specify.
>
> A high level description of the training pipeline for the entire framework is specified in the original paper in Chapter 3 and with more details Appendix A and B. Specifically, we use high-resolution real images to simulate lower resolution sensor output. This simulation step additionally depends on differentiable parameters that specify the sensor layout. Since standard networks cannot deal with non-uniform layouts, as they have explicit notion of distance between pixels, we reinterpret the non-uniform sensor simulation output as if it was on a uniform grid. This leads to a deformed image like the bird in Fig. 1., with areas of high pixel density being enlarged. This deformed image is fed into a task-dependent network, e.g. a segmentation network. Since the output of that network (the segmentation mask) is also deformed and has lower resolution, we resample it to be able to compare it to ground truth. Because every step in this pipeline is differentiable, we can propagate the gradients back to both the task-dependent network and the layout parameters. The code for this process will be made publicly available upon acceptance.
>
> Does this resolve your remaining concerns? We are happy to address any further questions. Please specify parts of our approach which remained unclear.

---

> > ### Comment · Reviewer_iE4U · 2023-11-21
> >
> > Thanks for the detailed feedback. Below are the remaining questions I have
> > * Is there any rationale behind choosing the smaller resolution size as the input rather than using the original size?
> > * If all the evaluation is conducted in simulation, how to verify the hardware design and what's the main take-away you want the reader to take from this?

---

> > > ### Author Response · Authors · 2023-11-22
> > >
> > > Thank you again for your time. We will gladly answer your remaining questions:
> > >
> > > >Is there any rationale behind choosing the smaller resolution size as the input rather than using the original size?
> > >
> > > An accurate sensor simulation requires multiple samples per pixel. In our derivations we assume a *continuous* radiance function, which in practice we approximate with a high-resolution image (e.g. 2048x1024 for the segmentation experiments), which we use as the input for our sensor simulation.
> > >
> > > Our overall objective is to achieve as high accuracy as possible with low pixel counts - and thus low computational cost. Therefore we put more pixels into regions where the downstream network requires more information. The image which we use as the input to the sensor simulation already has the finest granularity of spatial information that is available to us. As such, there is no additional information to gain from putting smaller pixels in some areas that are smaller than the pixels of the original image.
> > >
> > > >If all the evaluation is conducted in simulation, how to verify the hardware design and what's the main take-away you want the reader to take from this?
> > >
> > > Our simulated experiments show that non-uniform (but fixed) layouts can have an advantage over uniform layouts if a spatial bias is present in the data. To the best of our knowledge our paper is the first to utilize this inherent spatial bias **over the whole dataset**, as other works, e.g. (Marin et al., 2019) and (Jin et al., 2021),  have only considered per-image resampling patterns that require both the availability of high-resolution images and incur additional computational cost.
> > >
> > > The IC layouts resulting from our optimization process have been **formally verified** to confirm the manufacturability of these non-uniform sensors. In the section “Sensor Hardware” of Chapter 4 as well as Appendix E we provide examples of IC layouts and details on the design process of a hardware implementation of non-uniform sensor layouts. The sizes of the individual pixels of the designed layout are all in a region where manufacturability is very robust. We expect to have an operative prototype camera comprising the fabricated chip, functionality tests and camera integration in about 12-18 months from now.

---

### Author Response · Authors · 2023-11-15

We'd like to thank the reviewers for taking the time to assess our work in detail and provide valuable feedback that helped us improve our work. In particular, we were delighted to read that our task-specific fully differentiable sensor layout optimization framework is "an important research direction", and a "novel contribution" that "can be easily incorporated into existing deep learning models" and "can adapt to different types of camera-based tasks" with concepts that were "clearly explained at the correct level of detail" and "well presented and motivated". We feel honored by your positive feedback! Below we provide detailed answers to all open questions individually.

---

### Meta-Review · Area_Chair_CF7x · 2023-12-12

**Metareview:**

The paper observes that there is spatial bias on some vision task, for example, in autonomous driving, the sensor needs higher spatial resolution in some regions (horizontal region and side regions). Based on this observation, the paper proposes the sensor spatial layout optimization for specific tasks, like semantic segmentation.

Strengths: (1) The first to come up with end-to-end sensor layout optimization. (2) The formulation of the problem to be differentiable, e.g., the parameterized deformation.

Weaknesses: (1) No real sensor and all experiments are by simulations. It's better to have real hardware, even if the hardware is crappy. (2) The lack of considerations of real factors, which weakens the motivation: [Generalization in diverse applications] The fixed sensor layout doesn't allow flexible camera tilting angle. [Generalization in different scenes] Is it effective in datasets with anomalous scene (usually it’s dangerous to ignore the abnormal conditions in autonomous driving-like application) or in robotics tasks with both indoor and outdoor activities? [Imaging problem] Spatially-varying pixel size leads to spatially-varying noise level and spatially-varying dynamic range. [Training data] It doesn't enjoy the huge amount training data anymore because most data is from regular sensor layout. [Lens as an alternative] Lens design can achieve the same effect - spatially-varying resolution, and is much cheaper than sensor design, mostly because lens manufacturing is much cheaper than sensor design. Here the lens can also be a curved mirror. [Lens's geometric distortion] Current paper doesn't consider the lens' distortion. (3) The lack of comparisons: [Exp. on different input resolution] currently all experiments are at low resolution. [An alternative method] How about doing an statistics on objects’ spatial distribution and then just doing an one-step optimization on layout? (4) The paper makes more sense if the resolution can be adaptively changed, like + galvo mirror/liquid lens, but the paper didn’t mention it.

The paper got large spread in ratings (rating/confidence): 5/4, 8/5, 5/4, 8/3. While the reviewers and ACs appreciate the initial effort put into the sensor layout design, there are still some issues that have not been resolved, which is summarized in the Weaknesses part above.
Taking all factors into account, the area chair does not recommend this paper be accepted at ICLR because of lack of real experiments and consideration of practical issues (generalization in diverse applications and different scenes, imaging problem, training data resource, alternative methods, etc.).

**Justification For Why Not Higher Score:**

There are two major problems for the paper. (1) There is no real sensor and all experiments are by simulations. (2) The lack of considerations of practical factors (generalization in diverse applications and different scenes, imaging problem, training data resource, etc.) or simpler alternative methods, which makes the motivation of the paper weak.

**Justification For Why Not Lower Score:**

N/A

---

### Decision · Program_Chairs · 2024-01-16

Reject